# Genomic and Metabolic Characterization of Plant Growth-Promoting Rhizobacteria Isolated from Nodules of Clovers Grown in Non-Farmed Soil

**DOI:** 10.3390/ijms242316679

**Published:** 2023-11-23

**Authors:** Magdalena Wójcik, Piotr Koper, Kamil Żebracki, Małgorzata Marczak, Andrzej Mazur

**Affiliations:** Department of Genetics and Microbiology, Institute of Biological Sciences, Maria Curie-Skłodowska University, Akademicka 19 St., 20-033 Lublin, Poland; magdalena.wojcik2@mail.umcs.pl (M.W.); piotr.koper@mail.umcs.pl (P.K.); kamil.zebracki@mail.umcs.pl (K.Ż.); malgorzata.marczak@mail.umcs.pl (M.M.)

**Keywords:** plant microbiota, plant growth-promoting rhizobacteria (PGPR), *Rhizobium*, genomic characteristics, metabolic profiling

## Abstract

The rhizosphere microbiota, which includes plant growth-promoting rhizobacteria (PGPR), is essential for nutrient acquisition, protection against pathogens, and abiotic stress tolerance in plants. However, agricultural practices affect the composition and functions of microbiota, reducing their beneficial effects on plant growth and health. Among PGPR, rhizobia form mutually beneficial symbiosis with legumes. In this study, we characterized 16 clover nodule isolates from non-farmed soil to explore their plant growth-promoting (PGP) potential, hypothesizing that these bacteria may possess unique, unaltered PGP traits, compared to those affected by common agricultural practices. Biolog profiling revealed their versatile metabolic capabilities, enabling them to utilize a wide range of carbon and energy sources. All isolates were effective phosphate solubilizers, and individual strains exhibited 1-aminocyclopropane-1-carboxylate deaminase and metal ion chelation activities. Metabolically active strains showed improved performance in symbiotic interactions with plants. Comparative genomics revealed that the genomes of five nodule isolates contained a significantly enriched fraction of unique genes associated with quorum sensing and aromatic compound degradation. As the potential of PGPR in agriculture grows, we emphasize the importance of the molecular and metabolic characterization of PGP traits as a fundamental step towards their subsequent application in the field as an alternative to chemical fertilizers and supplements.

## 1. Introduction

Studies in recent decades have led to the discovery of complex communities of microorganisms associated with various plants and specific plant parts. These microorganisms, collectively referred to as the plant holobiont or plant microbiota, encompass all microorganisms present in the rhizosphere, phyllosphere, and endosphere of plants. The plant microbiota plays a key role in supporting plant growth and health [1], being responsible for nutrient acquisition, protection against pathogens, and tolerance to abiotic stress [2]. Understanding the interactions between plants and microbes, as well as the factors that shape microbial communities, can provide valuable insights into how these microorganisms contribute to plant growth and crop production [3].

The soil microbiota comprises diverse communities of bacteria, archaea, fungi, protozoa, and viruses [4]. It plays a pivotal role in maintaining soil health, nutrient cycling, plant growth, and overall ecosystem functioning [3]. Due to the high microbial content in the soil and the release of various metabolites from plant roots into the rhizosphere (known as rhizodeposition), the rhizosphere microbiota appears to play a special role in promoting plant growth. Rhizodeposition drives interactions between plants and microbial populations and controls many soil ecological functions, including nutrient availability and mobilization [5].

The processes of plant domestication and agricultural practices, including artificial fertilization, have had adverse effects on the composition and functions associated with plant microbiota. Research has shown that the profile of metabolites, mainly simple sugars, secreted by the roots of cultivated plants significantly differs from that of their wild counterparts [6]. This variance affects the composition of the rhizosphere microbiota, weakening its positive impact on plant growth and health.

Plant growth-promoting rhizobacteria (PGPR) are rhizosphere bacteria that can enhance plant growth through several direct and indirect mechanisms. Direct modes of action include PGPR’s abilities of biological nitrogen fixation (BNF), phosphate solubilization, siderophore and other metallophore production, and the production of ethylene and phytohormones. Indirect mechanisms involve the production of bioactive compounds (e.g., with potential as biocontrol agents against pathogens): antibiotics, volatile organic compounds (VOCs), and lytic enzymes, as well as the induction of plant systemic resistance [7,8,9]. Moreover, the indirect plant growth-promoting traits (PGPTs) of these bacteria, such as the biosynthesis of amines, the production of 1-aminocyclopropane-1-carboxylate (ACC) deaminase, antifreeze proteins, and antioxidant enzymes, provide stress tolerance to the host plant [9]. The potential of PGPR in agriculture is steadily increasing as it offers an attractive alternative to chemical fertilizers, pesticides, and other supplements [9]. Within the category of PGPR, soil bacteria, collectively referred to as rhizobia, form beneficial symbiosis with leguminous plants, which are vital for both human food and animal feed production [10]. Rhizobia induce the formation of nodules on the roots or stems of these plants and fix nitrogen from the air, providing it to the plant [11]. Rhizobia, as a significant component of the plant rhizosphere microbiota, not only make these plants independent of nitrogen fertilization but also contribute to the global nitrogen cycle [12]. Rhizobia thrive in soil environments with limited and diverse resources, competing for nutrients with other bacteria present in the host plant rhizosphere [13,14]. Rhizobial genomes are large and complex, reflecting their diverse metabolic capabilities [15]. The size of rhizobial genomes can range from 5 to 9 Mb [16]. Sequenced rhizobial genomes usually consist of a single circular chromosome, often accompanied by a set of extrachromosomal replicons (ECRs) with sizes ranging from several kb to Mb [16,17,18,19]. Such genomes are presumed to offer ecological advantages in challenging environments like soil. The accessory genome of rhizobia, comprising the set of ECR, frequently carries the *nod* and *nif*/*fix* genes responsible for the symbiotic interaction with legumes, as well as many other PGPTs [20]. Additionally, genes that enable rhizobia to adapt to various environments are located on ECRs. [21,22].

Our previous studies on the population of *Rhizobium leguminosarum* bv. *trifolii* (*Rlt*), a microsymbiont of clover, isolated from the root nodules of plants growing in the same soil site, revealed a significant divergence of their genome organization, particularly regarding their ECR content [23]. These isolates harbored between 3 and 6 ECRs, with sizes ranging from approximately 150 kb to 1380 kb. Biolog phenotypic profiling, including the utilization of various carbon, nitrogen, phosphorus, and sulfur sources, as well as tolerance to osmolytes and pH, demonstrated the metabolic versatility of the sampled *Rlt* strains. They could utilize various substrates, with notable preferences for sugar alcohols, glycosides, nucleosides, nucleotides, oligopeptides (mainly dipeptides), and monosaccharides. Some strains exhibited significantly diverse metabolisms and proved to be effective clover inoculants.

The main objective of this study was to characterize the population of nodule isolates obtained from white and red clovers growing in the buffer zone of Białowieża National Park (BNP), with respect to their plant growth-promoting (PGP) potential. Our hypothesis was that the symbiotic bacteria forming the rhizosphere microbiota of plants growing in non-farmed land may exhibit unique metabolic features related to their adaptation to growth and symbiosis, which remain unaltered by agricultural practices. Such bacteria, or their metabolically active genes, may hold the potential to promote plant growth.

## 2. Results

### 2.1. Microbiota Profiling of Clover Rhizo-Soil from Non-Farmed Land

The starting point of the analyses was the comparison of the rhizosphere soil microbiota composition of clover plants grown in non-farmed land versus agricultural land (previously treated with fertilizers). We aimed to verify whether any of these microbiota, particularly those from non-farmed soil, exhibited a significant enrichment in PGP taxa. Initially, a total of 810,474 input sequences were obtained for the tested samples. After preprocessing, quality filtering, and removing chimeras, 808,474 sequences remained. Out of these, 374,385 (46.2% of the input sequences) were assigned to operational taxonomic units (OTUs) and taxa. The total number of detected OTUs was 889, and all of them were accurately assigned to taxa (Appendix A).

At the phylum level, Acidobacteria, Actinobacteria, and Proteobacteria were the dominant phyla in the clover rhizosphere of both tested soil types, and their relative abundances did not show significant differences between the groups (Figure 1). The most prevalent genera included *Bradyrhizobium*, *Gaiella*, *Acidobacterium, Candidatus Koribacer*, and *Candidatus Solibacter*, among others. However, at the genus level, we did not observe a significantly higher content of bacteria belonging to the plant growth-promoting taxa (PGPB) in the rhizosphere of clovers grown in non-farmed soil.

Alpha diversity indexes (the Chao1 estimator, Shannon index, and Simpson diversity index), calculated based on 16S rRNA sequencing data, indicated high bacterial richness and diversity in the tested communities. However, no statistically significant differences between the analyzed groups were found (Appendix A). It may be concluded that agricultural practices did not have a significant effect on the composition of the rhizosphere microbiota, at least for the tested plant species.

Although we did not find substantial differences in the composition of the rhizosphere microbiota between the analyzed soils and could not confirm the prevalence of PGPB taxa in the rhizosphere of plants growing in the non-farmed area, *Rhizobium* isolates were obtained from the root nodules of clovers growing in the buffer zone of the BNP. We hypothesized that bacteria that are successful in colonizing plants may possess unique metabolic activities that are favorable to the host. A total of 16 strains were isolated from the nodules of the white and red clovers, designated as KB2–KB12 and KC2–KC8, respectively.

A macroscopic phenotype screening of the KB and KC strains revealed that most of them grew well on the standard media (79CA, M1, and TY), forming mucoid colonies, with only minor differences in growth kinetics and polysaccharide secretion (Appendix A). The initial taxonomic classification of the strains as *Rhizobium leguminosarum* was established through a sequence analyses of the 16S rRNA and *nodA* gene fragments. The sequence similarity of both analyzed gene fragments to the reference *Rlt* strains exceeded 99%. The isolates were subsequently genotyped to determine their ECR content using the Eckhardt technique and displayed varying architectures of their extrachromosomal genomes in terms of both the number (ranging from three ECR for KC2 to seven for the KB11 strains) and the size of the ECRs (ranging from 0.085 Mb for KB3 to 1.145 Mb for strains KC2, KC4, KC5, KC7) (Appendix A), suggesting broad metabolic potential.

### 2.2. Phenotypic and Symbiotic Profiling of R. leguminosarum Isolates

The next step of this research involved the metabolic profiling of the obtained *R. leguminosarum* isolates using Biolog GEN III MicroPlates. The utilization of 71 carbon sources and the sensitivity of the bacteria to 23 chemical stressors were tested. To gain insight into the metabolic activity, their metabolic profiles were compared with those of previously characterized symbionts of red clover grown in arable lands (designated as K2.9, K3.6, K4.15, and K5.4). These symbionts exhibited diverse genomic architectures, substantially varied metabolism, and were effective clover inoculants [23]. An analysis of the Biolog data revealed that strains KB7, KB8, KB10, KB12, KC4, and KC5 displayed versatile metabolic capabilities, enabling them to utilize a wide range of carbon and energy sources. Additionally, they demonstrated higher tolerance to stressors (Figure 2). These strains differed in their ECR content, suggesting that the variability in their extrachromosomal genomes may contribute to their heightened metabolic potential. Isolates KB7, KB8, KB10, KB12, KC4, and KC5 efficiently utilized most of the tested sugars, with a strong preference for sucrose, galactose, trehalose, gentiobiose, and fructose, as well as turanose, mannose, lactose, glucose, and pectins (Figure 2A). More than half of the organic acid compounds were utilized by these KB and KC strains (Figure 2B). Among the efficiently metabolized substrates were D- and L-malic acid, bromosuccinic acid, acetic and acetoacetic acids, gluconic acid, citric acid, and quinic acid. Amino acids were utilized to a lesser extent, but a subset of KB7, KB8, KB10, and KB12 isolates effectively metabolized L-alanine and L-histidine, while KC4 and KC5 metabolized L-aspartic and L-pyroglutamatic acids (Figure 2C). Among the substrates categorized as ‘others’, KB7, KB8, KB10, KB12, KC5, and, to a lesser extent, KC4 efficiently utilized sorbitol, mannitol, arabitol, glycerol, and myoinositol (Figure 2D). In the presence of stressors on the Biolog plates, the KB and KC strains exhibited varying levels of antibiotic tolerance. Notably, KC4 showed resilience to tetrazolium violet, sodium lactate, and 1% NaCl, while KC5 demonstrated resistance to potassium tellurite. Additionally, among the KB strains, KB8 and KB10 displayed some tolerance to pH 5, and it is worth noting that most of the KB and KC strains exhibited metabolic activity in the presence of tetrazolium blue (Figure 2E).

Individual strains were also subjected to biochemical tests to assess their capacity for phytohormone synthesis, metallophore production, and enhanced phosphorus bioavailability. Only the strains within the KC group exhibited ACC deaminase activity, enabling them to grow in the presence of ACC as the sole nitrogen source (Figure 3). Four metals were selected to screen metallophore production by the study strains and included Cu^2+^, Al^3+^, Fe^2+^, and As^3+^. A yellow-orange halo surrounding the bacterial colonies demonstrated the ability of the majority of the strains to chelate Cu^2+^ (Figure 3). For the other metal ions tested, only selected strains showed a moderate chelating capacity, visible as a color change within the colony, for Al^3+^ (KB3, KB6, KB7, KC3), Fe^2+^ (KB3, KB6, KB8, KC2) and As^3+^ (KB3, KB6, KB8) (Figure 3). The strains’ ability to solubilize tricalcium phosphate was assessed on Pikovskaya’s agar. All isolates formed visible dissolution zones and thus were identified as effective phosphate solubilizers (Figure 3).

The KB and KC strains displaying versatile metabolic profiles were analyzed in respect of their symbiotic performance in plant tests. The fresh mass of clover shoots inoculated with KC4 and KC5 was slightly higher than that of plants infected with the control strains K2.9 and K4.15, which were previously described as symbiotically active and effective nitrogen-fixers (Figure 4). The fresh mass of shoots from plants inoculated with KB3, KB7, KB8, KB10, and KB12 was significantly higher than that of non-inoculated plants, but lower than that of KC4 and KC5.

### 2.3. Genome Sequencing and Analysis

Five selected strains of *R. leguminosarum*, specifically KB7, KB8, KB12, KC4, and KC5, which exhibited the highest metabolic activity and symbiotic efficiency in previous tests, were subjected to complete genome sequencing and comparative genomics to identify the gene clusters responsible for the metabolic traits associated with PGP activity. The analyses yielded five high-quality draft genome sequences (Appendix A).

The genomes sequenced in this study showed remarkable similarity in their key characteristics (Appendix A). They displayed comparable sizes, ranging from 7.09 Mb (KB7) to 7.41 Mb (KB8 and KB12), and G+C contents, with values ranging from 60.72% (KB8 and KB12) to 60.79% (KB7). The genome assemblies were highly contiguous, with N50 values ranging from 221,519 bp (KB7) to 395,773 bp (KC5). The completeness estimates were uniformly high, with a value of 100% for all genomes, and the level of contamination was found to be negligible.

Genome annotations identified a similar number of coding sequences (CDSs), tRNAs, and rRNAs across all genomes, with approximately 7500 CDSs, 46 tRNAs, and three rRNAs per genome. Phylogenetic analyses, performed using the autoMLST v.1 tool, which very rigorously combines whole-genome similarity analyses with phylogenetic inference based on selected genetic markers, classified KB strains as *Rlt* and assigned KC strains to the taxonomic position of *Rhizobium gallicum* (Figure 5).

Based on the phylogenetic analysis, 12 complete genomes of *Rlt* strains available in the RefSeq database were selected, as well as three complete genomes of *R. gallicum* strains, including reference and type strains. All genomes were compared with the sequences obtained for KB and KC strains, resulting in the construction of a pangenome and the identification of common genes, as well as genes unique to individual isolates (Figure 6).

A comparative analysis of the constructed pangenome revealed a number of PGP trait-encoding genes among the common core genes. These included genes related to biofilm formation, nodulation, and nitrogen fixation, as well as those related to nitric oxide reduction, a lowering of the ethylene level, and auxin synthesis [24,25,26]. It is worth mentioning that the *acdS* gene was identified only in the KC group, which is in agreement with the results of ACC deaminase activity assay described above (Table 1).

The final pangenome matrix was used for downstream comparative genomics and functional enrichment analysis. Based on this matrix, gene clusters specific to the KB and KC genome groups were selected and subjected to functional enrichment using KEGG pathways.

Both KB and KC genomes were significantly enriched in functions related to quorum sensing and the degradation of aromatic compounds such as benzoate (and its derivatives), styrene, and xylene. In addition, KEGG pathways related to phenylalanine and tryptophan metabolism were significantly over-represented in strains of the KB group (Figure 7).

To further explore the functional enrichment in detail, enriched genes were mapped onto individual metabolic pathway maps provided by KEGG (Appendix A). This approach allowed us to show that downstream pathways dependent on quorum-sensing functions include those related to flagellar assembly (related to the *qseB* gene), conjugal plasmid transfer (*tra* genes), the synthesis of bacteriocins and immunity proteins (*blpA* and *blpB* genes), the maturation of antimicrobial peptides (*nisB* and *nisC* genes), virulence (related to the dipeptide permease *dpp* gene), and quorum quenching (*blcC* gene). These quorum-sensing-dependent features may allow the strains to compete with other microorganisms present in the rhizosphere and probably provide them with advantages contributing to successful infection of host plants. In turn, downstream pathways, in which products of aromatic compound degradation (particularly benzoate and styrene) can be used include citrate cycle (TCA), propanoate, and pyruvate metabolism, with the latter pathway being strictly related to symbiotic nitrogen fixation. The ability to convert aromatic compounds to simpler intermediates, which are further metabolized in basic cellular pathways, increases the metabolic flexibility of rhizobia and contributes to a better adaptation to their environment.

## 3. Discussion

The excessive use of agrochemicals in the field, aimed at increasing production and mitigating the negative effects of biotic and abiotic factors, has led to a decline in soil fertility. It has also exacerbated the negative effects on the environment and human health. Consequently, the use of beneficial microorganisms as bioinoculants represents an environmentally friendly alternative to agrochemicals. A wide range of plant growth-promoting bacteria and fungi have been shown to be effective in increasing plant growth and yield in a variety of crops [27]. The soil serves as a perfect source for the search for new PGP microorganisms. The soil holobiont exhibits remarkable diversity, a factor of great importance, as it contributes to the stability and robustness of soil ecosystems [27]. Soil microbiota are involved in the cycling of nutrients such as carbon, nitrogen, phosphorus, and sulfur. Soil microbes also play a critical role in the decomposition of organic matter, such as dead plants and animal residues [28]. They secrete enzymes that break down complex organic compounds into simpler forms, releasing nutrients that can be readily absorbed by plants. The process of decomposition and nutrient mineralization is essential for maintaining soil fertility and ensuring nutrient availability [29]. Some soil microorganisms have the ability to suppress plant pathogens and protect plants from disease [30]. They can compete with pathogens for resources, produce antimicrobial compounds, or induce systemic resistance in plants. Microbes produce polysaccharides and other substances that bind soil particles together, contributing to the formation and stability of soil aggregates. This is crucial for soil structure and water infiltration, and thus acts as a defense against erosion [31].

The soil microbiota is more abundant and complex in the rhizosphere, the narrow zone surrounding plant roots, with up to 10^9^ cells per gram in typical rhizospheric soil, comprising up to 10^6^ taxa [32]. Of particular interest in the rhizosphere are the plant growth-promoting rhizobacteria, which act through a variety of mechanisms [33]. Nitrogen-fixing bacteria, including free-living (e.g., *Azotobacter* spp.) and symbiotic (e.g., root-nodulating *Rhizobium* spp.) bacteria, provide a source of reduced nitrogen for the plant, and many bacteria can solubilize phosphorus-containing minerals, increasing their bioavailability [34]. The microbial modulation of plant hormones, such as auxins, gibberellins, and ethylene, can also result in growth promotion or stress tolerance. In addition, many PGP rhizobacteria can act antagonistically against plant pathogens by producing antimicrobials or interfering with virulence factors [34]. Unfortunately, the domestication of plant species and agricultural practices, such as the use of artificial fertilizers, have had detrimental effects on the delicate balance between plants and microorganisms, affecting the composition and functions of the plant microbiota [35].

### 3.1. Agricultural Practices Did Not Affect the Clover Rhizosphere Microbiota Composition

Considering the above factors, we hypothesized that the rhizobacteria present in the rhizosphere of plants growing in non-agricultural land may have unique metabolic traits related to their adaptation to growth and symbiosis. These features would remain unaltered by agricultural practices, and such bacteria, or their metabolically active genes, could potentially be used to enhance plant growth. We first assessed the composition, structure, and diversity of the microbiota in clover rhizo-soil using NGS-based 16S rRNA amplicon sequencing. This approach did not reveal substantial differences in microbial composition between the analyzed rhizo-soils. Previous studies of rhizosphere microbiomes have shown remarkably similar distributions of microbial phyla, while significant differences, e.g., between plant cultivars, became more apparent when microbial species and strains were compared [36,37]. The α and β classes of Proteobacteria usually predominate in such samples, with other major groups including Actinobacteria, Firmicutes, Bacteroidetes, Planctomycetes, Verrucomicrobia, and Acidobacteria. Actinomycetes have also been identified as one of the most abundant classes of bacteria in the soil and rhizosphere, and were particularly enriched in endophytic communities [34]. Representatives of these bacterial taxa were also identified in the clover rhizo-soils analyzed in this study. It is worth noting that root exudates play a pivotal role in shaping the microbiome structure of the rhizosphere [38,39]. The composition of root exudates can vary between plant species and cultivars [40], as well as with plant age and developmental stage [41]. Importantly, the activity of the microbiome influences root exudates, as axenically grown (sterile) plants have different root exudate compositions than plants influenced by microbes [34]. In our analyses, we did not confirm the significant prevalence of PGPB taxa in the rhizosphere of plants growing in the non-farmed land. However, other studies have demonstrated that the rhizosphere microbiome of native plants can serve as an excellent resource for plant growth-promoting rhizobacteria, which can be utilized as biofertilizers and biostimulants [42].

### 3.2. The Clover Nodule Bacteria Display Potentially Useful PGT

By focusing on bacteria that successfully colonized white and red clover nodules, we obtained a set of 16 strains with diverse metabolic properties. There is increasing evidence that plants support below-ground microbial biodiversity, mainly through root exudation and the process of rhizodeposition [43,44]. The rhizosphere is known to be strongly influenced by root exudates, mucilage, and exfoliated cells. It is therefore expected that the metabolic preferences of bacteria that successfully invade plants will match the profile of the metabolites found in the rhizosphere. Root exudates themselves contain a variety of compounds, mainly sugars and organic acids, but also amino acids, fatty acids, vitamins, growth factors, hormones, and antimicrobial compounds [45]. The KB and KC isolates efficiently utilized most of the tested sugars, showing a strong preference for sucrose, galactose, trehalose, gentiobiose, and fructose, as well as turanose, mannose, lactose, and glucose. Simple carbohydrates released into the soil serve primarily as a food source for bacteria [46]. In addition, the KB and KC strains effectively utilized polysaccharides, such as pectins. It is important to note that root exudates are not the sole component of rhizodeposition. The sloughing of root cells and the release of mucilage leads to the deposition of plant cell wall polymers such as cellulose and pectin in the rhizosphere. The degradation of pectin releases methanol, and methanol metabolism in the rhizosphere has been demonstrated [47]. KB and KC strains also efficiently utilized sorbitol, mannitol, arabitol, glycerol, myoinositol, and numerous organic compounds, while amino acids were utilized to a lesser extent.

Among the PGP traits of the studied strains, effective phosphate solubilization activity was demonstrated in all isolates. Phosphorus is often considered the second most important nutrient for plant growth and productivity after nitrogen [48]. Phosphorus availability in soils is often limited because it binds strongly to soil particles and divalent cations, forming insoluble phosphate complexes. Therefore, there is a constant need to improve soil phosphorus availability in a sustainable manner, for example, through the application of phosphate-solubilizing bacteria that are efficient in cycling this nutrient. This underlines the importance of microbial screening to ensure their effective selection [49].

The isolates in the KC group exhibited the activity of ACC deaminase, an enzyme that reduces the amount of ethylene produced by plants in response to stress. The ability of bacteria to degrade plant hormones or hormone precursors is also thought to promote plant growth, and the microbial deamination of ACC prevents plant ethylene signaling, making plants more tolerant to environmental stress [50]. In addition, individual strains have shown the ability to chelate metal ions, which was particularly evident for Cu^2+^. Bacteria, which produce metallophores to bind, for example, limiting Fe^3+^ in the soil, are an important category of plant growth-promoting rhizobacteria and have been shown to play vital roles in disease prevention and plant growth enhancement [51].

### 3.3. The Genomes of Clover Nodule Isolates Are Enriched in the Metabolically Adaptive Genes

In this study, the sequencing and comparative analysis of the genomes of selected KB and KC isolates revealed a substantial presence of PGP genes within the core gene set of these bacteria. These genes include functions related to biofilm formation, nodulation, and nitrogen fixation, as well as processes such as nitric oxide reduction, ethylene level regulation, and auxin synthesis. Notably, the functional enrichment of distinct gene clusters unique to KB and KC, when assessed against KEGG pathways, highlighted their specialized metabolic capabilities. Both KB and KC genomes exhibited significant enrichment in functions related to quorum sensing and aromatic compound degradation. Moreover, the KB strains showed a significant over-representation of KEGG pathways related to phenylalanine and tryptophan metabolism. Several downstream pathways potentially dependent on quorum-sensing functions, described in the Results section, deserve further attention. One of these pathways is related to flagellar assembly and the putative *qseB/qseC* genes. QseB/QseC is a two-component system involved in the regulation of many bacterial behaviors. In *Escherichia coli*, QseB/QseC signaling regulates the expression of more than 50 genes encoding flagellar proteins and proteins associated with virulence [52]. While QseB/QseC primarily regulates the bacterial infection processes of pathogens, it also confers survival advantages to environmental bacteria by providing resistance to host immune responses, flagella-driven motility, and antibiotic resistance [53]. These capabilities may provide rhizobacteria with an advantage that contributes to the successful colonization and infection of host plants. Another mechanism that is potentially dependent on the over-representation of quorum-sensing functions in tested strains is the production of bacteriocin-like proteins. The *blp* (bacteriocin-like protein) locus of *Streptococcus pneumoniae* has been shown to be responsible for bacteriocin production and immunity. Alterations in *blp* locus gene content have been shown to play an important role in competitive interactions between pneumococcal strains [54]. The production of putative antimicrobial compounds may allow bacteria to compete with other microorganisms present in the environment. Closely related to this potentially advantageous mechanism is another involving the activation of genes responsible for the maturation of antimicrobial peptides. In our pangenome analysis pipeline, this mechanism was mapped to KEGG pathways involving *nisB* and *nisC,* which are involved in the secretion of nisin precursors in several *Lactococcus lactis* strains [55].

Among the genes/functions that may be further dependent on quorum sensing in the KB and KC strains, the dipeptide permease *dpp* gene was also identified. The expression of the *R. leguminosarum* dipeptide transporter gene *dppA3* was strongly upregulated in bean and pea bacteroids, which are symbiotic forms of rhizobia. The mutation of the rhizobial *dpp* operon reduces the uptake of the heme precursor δ-aminolevulinic acid [56]. An essential determinant that nitrogen fixation in the *Rhizobium*–legume symbiosis is possible is leghemoglobin, a myoglobin-like hemoprotein that regulates the oxygen level in the nodule, thereby protecting the oxygen-sensitive nitrogenase from inactivation. While apoleghemoglobin is a plant gene product, the heme prosthetic group of leghemoglobin appears to be produced by the differentiated bacteria or bacteroids and contributes to the effective nitrogen fixation process [57]. In *Sinorhizobium meliloti*, a chromosomally encoded dipeptide permease (Dpp1) is required for the utilization of dipeptides and tripeptides, with a minor role for the utilization of tetrapeptides. In *Rhizobium etli*, an oligopeptide ABC transporter (Opt) mutant showed a significant reduction in symbiotic nitrogen fixation activity [58]. Importantly, before establishing a symbiosis with legumes, rhizobia must thrive in the soil and compete with many organisms for nutrients. Nutrient transporters may be advantageous for rhizobia, allowing them to colonize roots competitively. This may explain the large number of ABC transporters encoded in rhizobial genomes [59,60,61].

We have found that the KB and KC genomes were also significantly enriched in genes related to the degradation of aromatic compounds, including benzoate and its derivatives, styrene, and xylene. The downstream metabolic pathways, in which the products of the degradation of these aromatic compounds may be used, included the citrate cycle, propanoate, and pyruvate metabolism. The latter pathway is closely linked to symbiotic nitrogen fixation, which is energized via the metabolism of dicarboxylic acids, which requires their oxidation to both oxaloacetate and pyruvate [62]. There is a complex relationship between pyruvate synthesis in bacteroids, nitrogen fixation, and plant growth, generally associated with the promotion of ammonia secretion (derived from fixed nitrogen) into the plants [63,64]. Aromatic compounds are highly abundant in the soil and they can serve as a normal carbon source. The ability to convert structurally diverse aromatic compounds to structurally simpler intermediates, which are metabolized to tricarboxylic acid intermediates, provides great metabolic flexibility and contributes to the increased adaptation of bacteria to their environment [65]. The ability of rhizobia to degrade organic compounds, including pollutants, and their resistance to heavy metals have been previously reported, making them promising candidates for soil remediation in contaminated areas. It has also been demonstrated that rhizobia can synergistically stimulate the survival and activity of other biodegrading bacteria, thereby reducing the concentration of contaminants [66].

## 4. Materials and Methods

### 4.1. Rhizosphere Soil Sampling

At the end of the 2020 growing season (October 2020), composite samples consisting of both soil and plant biomass of *Trifolium pratense* L. and *Trifolium repens* L. growing in the buffer zone of the Bialowieza National Park and on arable soils were collected. With great care, clods of soil were carefully removed from the roots. Individual roots were placed on paper, and soil material more than 2 mm from the root surface was removed. The remaining soil material that covered the plant roots to within 2 mm of the root surface was then removed. This soil layer is referred to as the rhizospheric soil. Each soil sample was homogenized (larger particles were crushed). The samples were cooled down to a temperature of 4 °C. Total DNA was isolated from the samples prepared in this manner.

### 4.2. Soil Microbiota Profiling: DNA Isolation, 16S rDNA Sequencing, and Data Processing

Total DNA was extracted from the clover rhizo-soil samples with the GeneMATRIX Soil DNA Purification Kit (EURx Sp. z o.o., Gdańsk, Poland). The quantity and quality of gDNA were checked spectrophotometrically (Synergy H1 reader, Agilent Technologies, Inc., Santa Clara, CA, USA) and in agarose gel electrophoresis. PCR amplification of the targeted regions (highly variable region of bacterial 16S rRNA gene V3–V4) was performed using primers 341F (5′-GGACTACNNGGGTATCTAAT-3′) and 806R (5′-GTGCCAGCMGCCGCGGTAA-3′), connected with barcodes. PCR products of an appropriate size were selected via 2% agarose gel electrophoresis. The same amount of PCR products from each sample were pooled, end-repaired, A-tailed, and further ligated with Illumina adapters. Libraries were sequenced on a paired-end Illumina platform to generate 250 bp paired-end raw reads (Raw PEs). Raw PEs were assigned to samples based on their unique barcodes and truncated by cutting off the barcode and primer sequences. Paired-end reads were merged using FLASH (V1.2.7) [67]. Quality filtering on the raw tags was performed under specific filtering conditions to obtain high-quality clean tags [68] according to QIIME (V1.7.0) [69]. The tags were compared with the reference database (SILVA138 database, www.arb-silva.de (accessed on 24 June 2021)) using the UCHIME algorithm [70] to detect chimera sequences, and then the chimera sequences were removed [71]. Those sequences were used for subsequent analysis. The summarizations obtained in each step of the data processing are shown in Appendix A. The obtained metagenomic sequences have been deposited at DDBJ/EMBL/GenBank under the accession KIBY00000000.

### 4.3. Bioinformatic Analysis of Metagenomic Sequencing Data

In order to study the microbial community composition in each sample, sequence analyses were performed with the Uparse software (Uparse v7.0.1090) [70] using sequences after chimera removal. OTUs (operational taxonomy units) were obtained via clustering with 97% identity on the effective tags of all samples. The representative sequence for each OTU was screened for further annotation. For each representative sequence, QIIME (Version 1.7.0) [72] with the Mothur method was performed against the SSUrRNA database of the SILVA138 database [73] for species annotation at each taxonomic rank (kingdom, phylum, class, order, family, genus, and species) (threshold: 0.8–1) [74]. The OTUs’ abundance information was normalized using a standard sequence number corresponding to the sample with the least sequences. A *t*-test was performed to determine bacterial species with significant variation between groups (*p* < 0.05) at various taxon ranks. Subsequent analyses of alpha- and beta-diversity were all performed on this output normalized data. The alpha-diversity for each sample, including Chao1, Shannon, and Simpson, was calculated with QIIME (Version 1.7.0) and displayed with the R software (Version 2.15.3). A beta-diversity analysis was used to evaluate differences of samples in species complexity; the beta-diversity on both weighted and unweighted UniFrac was calculated with the QIIME software (Version 1.7.0).

### 4.4. Isolation of Rhizobia from Clover Nodules, Morphological Analyses, and Taxonomic Assessment

The fresh nodules were dissected from the roots, rinsed thoroughly in water, surface sterilized via immersion in 0.1% HgCl_2_ for 1 min and in 75% ethanol for 1 min, and rinsed in sterile water. The nodules were individually crushed, material from the nodules was streaked onto the surface of 79CA plates, and the plates were incubated at 28 °C for 5–7 days [75]. Individual colonies were selected and their purity was checked through repeated striking of single colonies on 79CA medium.

To assess the colony’s morphology strains obtained from the clover plants, the nodules were grown in TY (tryptone–yeast extract–calcium chloride) [76], M1 [77], or 79CA with 1% mannitol or glycerol at 28 °C [78]. The isolates were further genotyped in respect of their plasmid content using the Eckhardt technique [79]. The strains were grouped according to the size and number of plasmids. The taxonomic assignment of the strains into *R. leguminosarum* species was based on sequence analyses of the 16S rRNA and *nodA* gene fragments amplified with the primers Y1 (5′-TGGCTCAGAACGAACGCTGGCGGC-3′), Y2 (5′-CCCACTGCTGCCTCCCGTAGGAGT-3′), [80] nodA-1 (5′-TGCRGTGGAARNTRNNCTGG-3′), and nodA-2 (5′-GGNCCGTCRTCRAAWGTCAR-3′) [81], respectively. The cycling conditions were as follows: 16S rRNA gene (95 °C/45 s, 60 °C/45 s, 72 °C/120 s) × 30 cycles and *nodA* gene (95 °C/45 s, 52 °C/60 s, 72 °C/120 s) × 35 cycles. The primers used in this work were synthesized at Genomed S. A. (Warsaw, Poland). Genomic DNA was isolated with the Bacterial & Yeast Genomic DNA Purification Kit (EURx Sp. z o.o., Gdańsk, Poland), according to the manufacturer’s protocol. PCR was performed with the high-fidelity Platinum SuperFi II DNA Polymerase (Thermo Fisher Scientific, Waltham, MA, USA) according to the manufacturer’s recommendations. Sanger DNA sequencing was performed in Genomed S. A. (Warsaw, Poland). The sequences were analyzed with the BLAST tool 2.9.0 [72] with the default settings.

### 4.5. Metabolic Profiling of R. leguminosarum Isolates

Metabolic profiling of the obtained *R. leguminosarum* isolates was performed using GEN III MicroPlates (Biolog, Inc., Hayward, CA, USA). The tested isolates (KB3, KB5, KB6, KB7, KB8, KB10, KB11, KB12, KC2, KC3, KC4, KC5, KC6, KC7, and KC8) and control ones (K2.9, K3.6, K4.15, and K.5.4 [23]) were cultured in the TY liquid medium for 24 h with shaking (180 RPM). Then, 1 mL was taken from each culture, centrifuged (5 min, 10,000 RCF), and rinsed in a saline solution. The supernatant was removed and the bacteria were suspended to OD_600_ of 0.01 in inoculating fluid. Then, the cell suspension was inoculated into the GEN III MicroPlate test plates, 100 µL per well, and subsequently incubated to allow the phenotypic fingerprint to be formed. Increased respiration causes a reduction of the tetrazolium redox dye, forming a purple color. Negative wells remain colorless, as does the negative control (A1) with no carbon source. There was also a positive control well (A10) used as a reference for the chemical sensitivity assay in columns 10–12.

### 4.6. Plant Tests

Seeds of red clover (*T. pratense* L. cv. Nike) and white clover (*T. repens* L. cv. Lipollo) were surface-sterilized and germinated in Petri dishes with a nitrogen-free plant medium [82]. Two-day-old seedlings were transferred to agar slants and left to grow the first cotyledons over a 4–5 day period. Subsequently, the plants were inoculated with 0.1 mL of bacterial suspension with an OD_600_ of 0.1 (KB3, KB7, KB8, KB10, KB12, KC4, KC5, K2.9, K4.15). These plants were cultivated in a greenhouse with a 14/10 h day/night cycle. After 6 weeks, the plants were harvested, and the masses of fresh shoots and roots were measured. The number of root nodules formed was monitored weekly for 6 weeks following plant inoculation. The experiment was conducted with 30 replicates for each strain. Significant differences among experimental variants were assessed using a one-way analysis of variance (ANOVA), and pairwise comparisons of means were conducted using the Tukey test (*p* < 0.05).

### 4.7. Genome Sequencing and Analysis

The starting point for this research was the isolation of high-quality genomic DNA from the selected strains. For this purpose, the bacteria were propagated in the 79CA liquid medium at 28 °C with shaking. Total DNA was isolated from liquid cultures using the Bacterial & Yeast Genomic DNA Purification Kit (EURx Sp. z o.o., Gdańsk, Poland), according to the manufacturer’s protocol. The quantity and quality of gDNA was checked spectrophotometrically (Synergy H1 reader, Agilent Technologies, Inc., Santa Clara, CA, USA), fluorometrically (Qubit 2.0 Fluorometer with Qubit dsDNA BR Assay Kit, Thermo Fisher Scientific, Waltham, MA, USA), and through agarose gel electrophoresis. DNA sequencing was performed on a NovaSeq 6000 machine. For each isolate, 5 to 8 million paired-end Illumina reads (2 × 150 bp) were obtained, resulting in genome coverage > 100×, which is the recognized standard for bacterial genomes.

The analysis of the bacterial genomes involved a series of bioinformatic tools and methodologies. Raw sequence reads were first subjected to quality assessment using FastQC 0.11.9 [83], followed by the trimming of low-quality sequences and adapter removal using Trimmomatic (v0.39) [84]. Subsequently, a genome assembly was performed using the Unicycler assembler (v0.4.4) [85]. Contig order refinement was carried out using RagTag (v2.1.0) [86] to enhance the accuracy of the assemblies.

Genome annotation and functional assignment, including the identification of protein-coding sequences (CDSs), rRNA, tRNA genes, and functional categorizations, were accomplished using both the Prokaryotic Genome Annotation Pipeline (PGAP) [87] and Rapid Annotations using Subsystems Technology (RAST) [88]. The resulting annotations were merged for comprehensive analysis. To assess assembly quality, the QUAST (v5.0.2) [89] was employed. Additionally, the completeness and contamination of genome annotations were rigorously evaluated with CheckM (v1.1.0) [90]. The automated multi-locus species tree (autoMLST) pipeline was used to generate a phylogenomic tree and establish the taxonomic position of individual isolates. The nucleotide sequences have been deposited at DDBJ/EMBL/GenBank under the following accessions: JAWJWG000000000 (KB7), JAWJWH000000000 (KB8), JAWJWI000000000 (KB12), JAWJWJ000000000 (KC4), and JAWJWK000000000 (KC5). The collective data are available under the BioProject accession number PRJNA1026475.

A pangenome analysis was conducted using the Anvi’o v7.1 [91] software suite. This analysis encompassed the strain under investigation and 12 *Rhizobium leguminosarum* bv. *Trifolii,* as well as three *Rhizobium gallicum* complete genomes downloaded from RefSeq, facilitating the exploration of genetic diversity, the identification of shared and unique genomic elements, and the elucidation of core and accessory genes within the bacterial population. The final pangenome matrix, which records the presence/absence of gene clusters across all isolates, was utilized for downstream comparative genomics and functional enrichment analysis. In addition to the pangenome analysis conducted, an ANIb analysis was performed for genome comparison, all within the Anvi’o framework.

Based on this matrix, gene clusters specific to the KB genome group as well as to the KC group were selected. From each list of clusters, unique KEGG identifiers (KOs) were extracted as a starting point for further functional enrichment analysis. The resultant KEGG identifier lists were subjected to an enrichment analysis using the cluster Profiler R 4.10.0 package [92]. Significantly enriched KEGG pathways were identified based on a hypergeometric test with a false discovery rate (FDR) correction for multiple testing. Only pathways with an FDR-corrected *p*-value below a predefined significance threshold (typically 0.05) were considered biologically meaningful.

The enrichment results were visualized using bar plots, providing a comprehensive overview of the enriched KEGG pathways and their interconnections. Bar plots displayed the top enriched pathways, with the height of the bars indicating the significance of enrichment.

Furthermore, to explore the functional enrichment at a more specific level, enriched genes were mapped onto individual metabolic pathway maps provided by KEGG. Default parameters were used for these tools if not mentioned specifically.

### 4.8. Screening for ACC Deaminase-Producing Bacteria

The activity of ACC deaminase was checked using a screening method of culturing bacteria on a medium with the addition of ACC as the only nitrogen source. The tested isolates were cultured in an M1 minimal medium in a volume of 2 mL for 24 h at 200 RPM. After this time, the bacteria were diluted to OD_600_ of 0.1 in a M1 minimal medium. Then, 1 mL of the culture was centrifuged at 10,000 RCF for 5 min, the supernatant was removed, and the bacteria were washed with 1 mL of saline solution. This step was repeated 3 times. Then, 2.5 μL were spotted on the surface of the Dworkin and Foster (DF) medium [93]. The inoculated plates were incubated at 28 °C for 7 days and the growth on the plates was checked daily. Strains that grow on the DF medium utilize ACC as the only source of nitrogen, a trait that is a consequence of the presence of the activity of the enzyme, ACC deaminase.

### 4.9. Phosphorus Solubilization 

Phosphorus solubilization provides a quantitative measure of the ability of microorganisms to convert insoluble phosphorus sources into soluble forms. The Pikovskaya medium was used to qualitatively assess the ability of bacterial isolates to solubilize phosphate [94]. The final pH was adjusted to 7.00 ± 0.02. The strains were inoculated onto the solid agar plates and the plates were incubated at 28 °C for 10 days. The experiment was performed twice using three plates each time. Phosphate solubilization was demonstrated by strains surrounded by a clear zone (P-solubilization area) [95].

### 4.10. Metallophore Production

The production of siderophores was studied using Chrome Azurol S (CAS) agar media as described by Schwyn and Neilands [96]. To study the production of other metallophores, a modified CAS test was applied [97]. In this case, the CAS-Fe-HDTMA solution was prepared by replacing FeCl_3_ × 6H_2_O with the appropriate metal salt, i.e., CuSO_4_ for Cu^2+^, AlCl_3_ × 6H_2_O for Al^3+^, and NaHAsO_4_ × 7H_2_O for As^3+^, to a final concentration equal to that for Fe^3+^. CAS agar plates were inoculated with bacterial isolate and incubated at 30 °C for 7 days. Colonies showing yellow-orange and purple color changes within and around them were considered as siderophore/metallophore-positive [98].

## 5. Conclusions

The use of plant growth-promoting rhizobacteria as crop inoculants plays an important role in modern agricultural production. Numerous studies have previously demonstrated that individual microorganisms can exert beneficial effects on plants, although the field performance of such individual PGPR species may be limited due to various factors. In this study, some PGP characteristics of rhizobia from non-agricultural soils were elucidated. Although we did not find a substantial effect of agricultural practices on the structure of the clover rhizosphere microbiota, the non-farmed soil itself or plants growing in such soil may be a good source of potentially useful microorganisms that can be used as part of microbial consortia, where additive or synergistic PGP effects are expected. The molecular characterization of PGP traits, genetic engineering, and subsequent incorporation into selected effective rhizobacterial isolates or whole bacterial rhizosphere communities may represent a powerful strategy to generate improved PGPR tailored for specific agricultural applications.

## Figures and Tables

**Figure 1 ijms-24-16679-f001:**
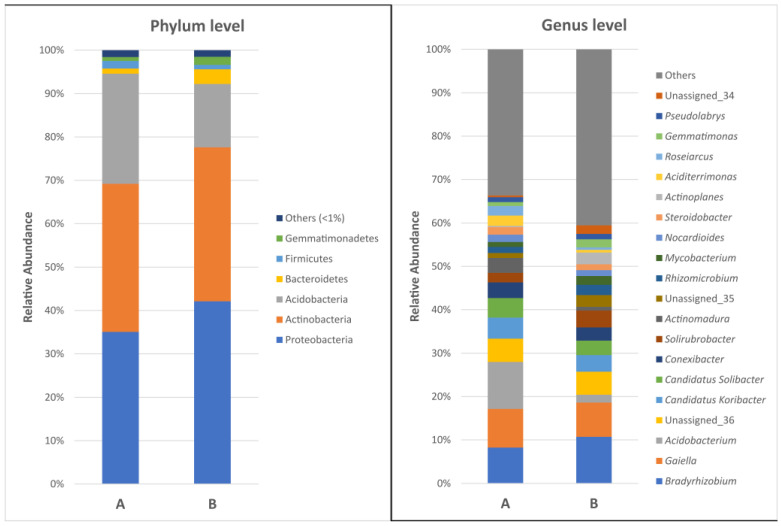
Taxonomic classification of bacterial communities at the phylum and the genus level in soil rhizosphere of clovers grown in non-farmed (A) and agricultural soil (B).

**Figure 2 ijms-24-16679-f002:**
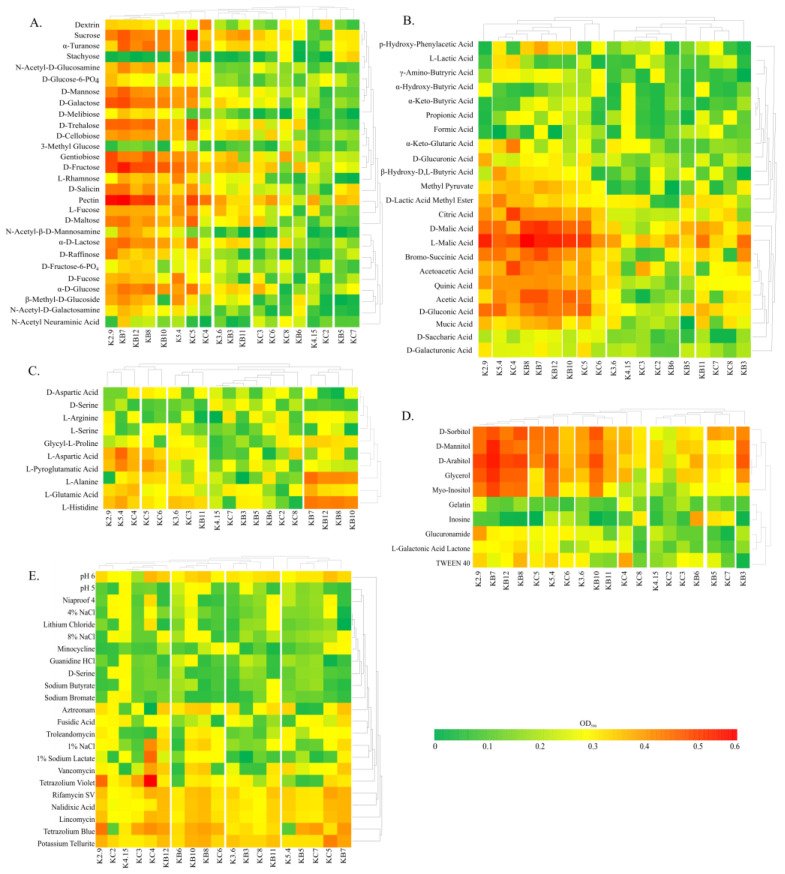
Heat maps of carbon utilization patterns of 71 substrates ((**A**)—carbohydrates and their derivatives, (**B**)—organic acids and their derivatives, (**C**)—amino acids, (**D**)—other compounds) and sensitivity to 23 stressors (**E**) from Biolog GEN III arrays used for the metabolic profiling of the obtained isolates. The results are presented as standardized data of absorbance measurement at 590 nm, where higher values indicate greater functional activity (**A**–**D**) or an increased tolerance level (**E**). The control group comprised previously characterized symbionts of red clover that grew in arable lands (K2.9, K3.6, K4.15, and K5.4).

**Figure 3 ijms-24-16679-f003:**
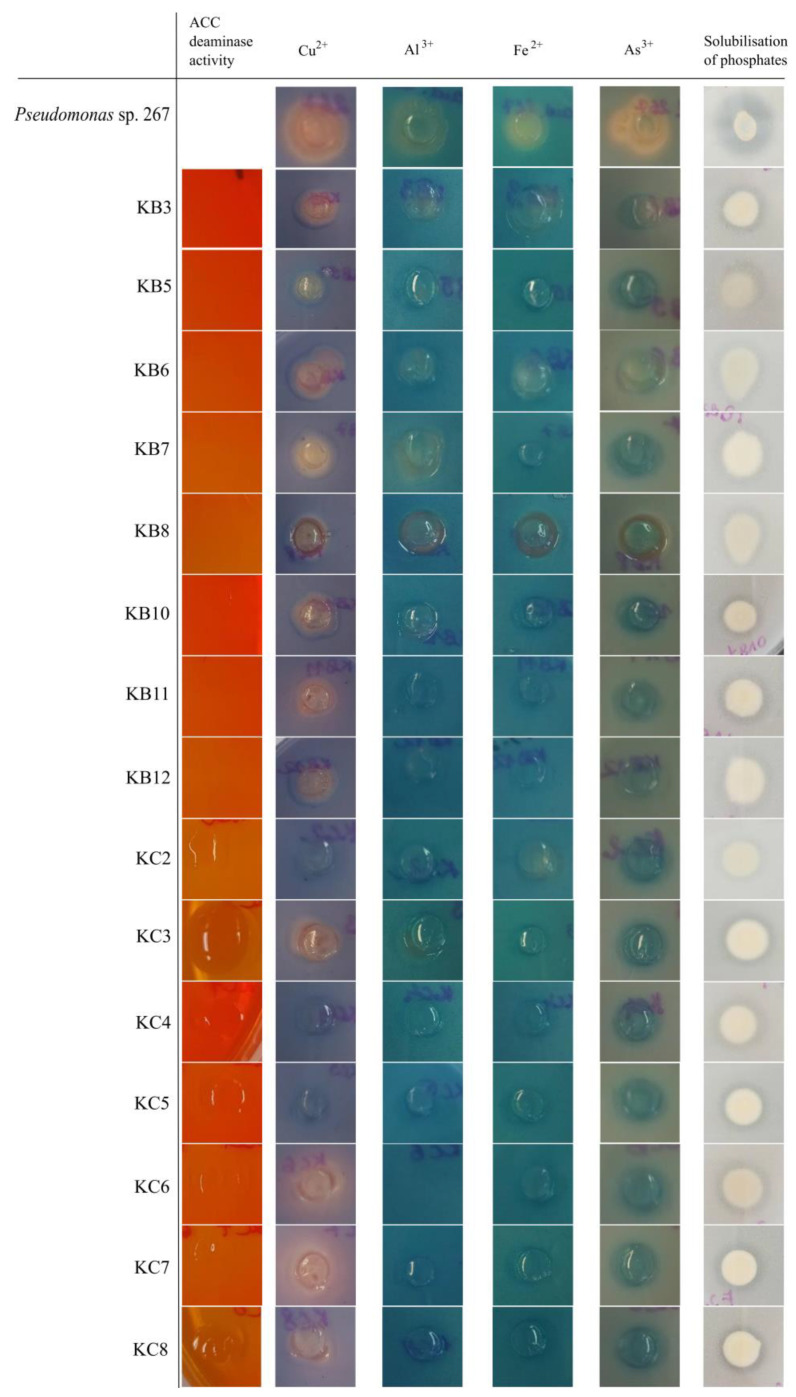
ACC deaminase synthesis, metallophore production, and phosphate solubilization activity of *R. leguminosarum* KB and KC isolates. ACC deaminase synthesis is indicated by growth in the medium containing ACC as the sole nitrogen source. Chelation of metal ions is visible as color changes to yellow-orange and purple in and around the bacterial colonies. Tricalcium phosphate dissolution zones around bacterial colonies point out phosphate solubilization properties.

**Figure 4 ijms-24-16679-f004:**
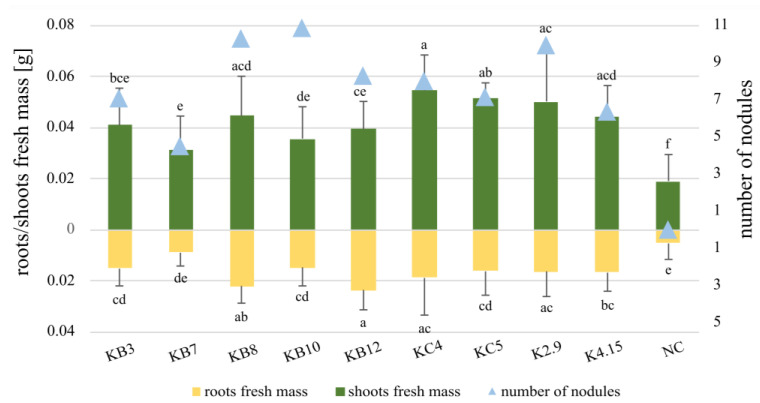
Symbiotic performance of *R. leguminosarum* KB and KC isolates. Symbiotic performance was assessed by measuring the fresh masses of shoots (depicted as green bars) and roots (depicted as yellow bars), and the number of induced nodules (depicted as blue triangles). The bars represent the mean values for 10 groups, each consisting of 30 plants: 7 groups were inoculated with the different strains under study (KB3, KB7, KB8, KB10, KB12, KC4, and KC5), 2 groups were inoculated with K2.9 and K4.15 as positive controls, and 1 group served as the negative control (NC) with non-inoculated plants. Extended black segments represent standard deviation. Statistical analyses were conducted using ANOVA and post hoc Tukey tests. Bars labeled with the same lowercase letters represent values with no significant differences, while those marked with different lowercase letters indicate mean values that exhibit statistically significant differences at *p* < 0.05.

**Figure 5 ijms-24-16679-f005:**
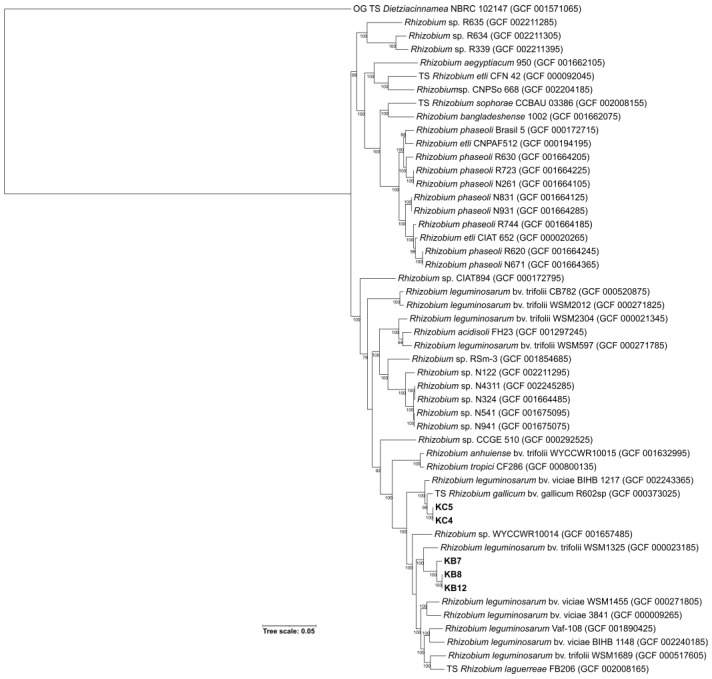
Maximum-likelihood phylogenomic tree based on 70 concatenated core genes, constructed using the autoMLST tool, which automates the extraction and analysis of multilocus sequence typing (MLST) data from genomic sequences. The tree includes both query strains (QSs) and type strains (TSs), which serve as references. *Dietzia cinnamea* NBRC 102147 was used as an outgroup (OG) to provide context for the diversity of *Rhizobium* species. The GenBank genome assembly accession numbers are shown in brackets. Bootstrap values greater than 70 are indicated in the nodes. The bar indicates sequence divergence.

**Figure 6 ijms-24-16679-f006:**
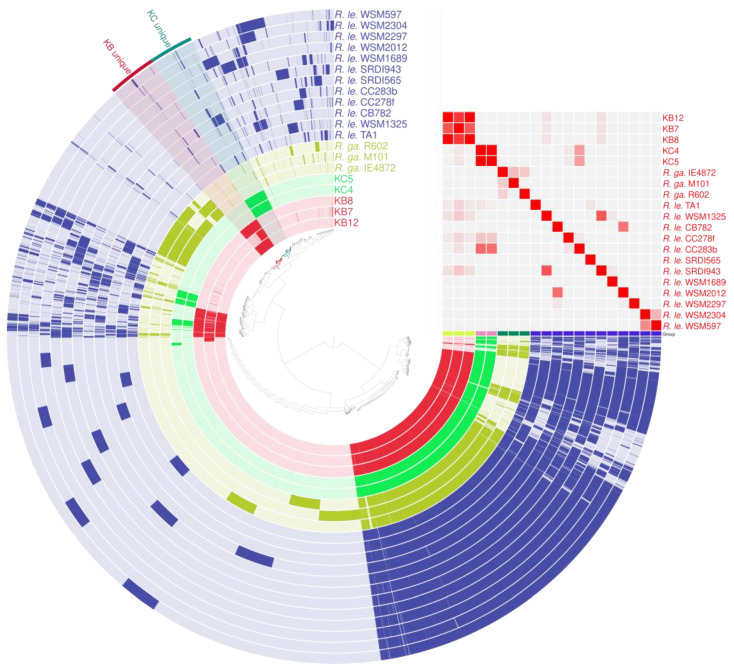
Pangenome analysis of KB, KC, and reference strains of *Rlt* and *R. gallicum*. Predicted protein sequences from the CDSs of all genomes analyzed were grouped into orthologous gene clusters. The presence of genes belonging to a particular cluster in a given genome is indicated with a colored bar. Each ring represents an individual genome. Genomes of strains derived from white clover (group KB) are marked in red and from red clover (group KC) in green. In addition, clusters of genes that are unique to the KB, KC, and both groups, were highlighted. The similarity matrix on the right shows the average nucleotide identity (ANI) comparisons between the analyzed genomes, providing an insight into the genetic relatedness between them, where the color scale reflects a similarity level ranging from 70 (grey) to 100% (red).

**Figure 7 ijms-24-16679-f007:**
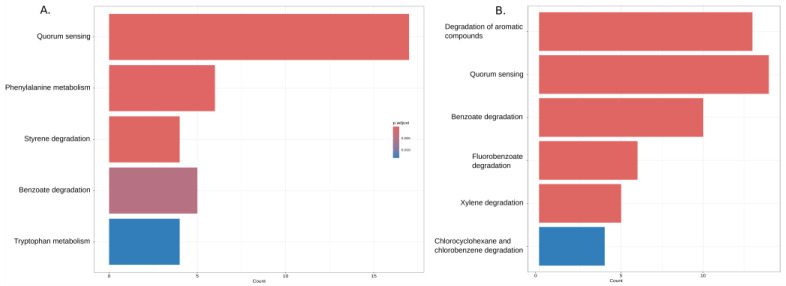
KEGG enrichment analysis results of KB (**A**) and KC (**B**) unique gene clusters. The bar graph illustrates the KEGG pathway enrichment analysis results for a set of genes. Each bar represents a KEGG pathway category, and the color intensity corresponds to the enrichment significance level (*p*-value). Reddish colors indicate more significant enrichment, highlighting pathways of potential biological importance.

**Table 1 ijms-24-16679-t001:** Distribution of genes contributing to plant-beneficial functions identified in the genomes of the KB and KC groups compared with two reference strains, *R. gallicum* M101 and *Rlt* WSM1325. (+/−) means presence or absence of the gene in the KB/KC strains.

		Strain						
Gene Function	Gene	*R. gallicum* M101	*Rlt* WSM1325	KB7	KB8	KB12	KC4	KC5
Biofilm formation	*bolA*	+	+	+	+	+	+	+
	*cvpA*	+	+	+	+	+	+	+
Lowering the ethylene level	*luxI (rail)*	+	+	+	+	+	−	−
Nodulation	*nodA*	+	+	+	+	+	+	+
	*nodS*	+	+	+	+	+	+	+
	*pqqC*	+	−	+	+	+	−	−
Nitric oxide reductase	*norD*	+	+	+	+	+	−	−
	*norQ*	−	−	−	+	+	−	−
Nitrate reduction	*nirD*	+	+	+	+	+	+	+
Auxin synthesis	*ipdC*	+	+	−	−	−	+	+
	*yfdV*	+	+	+	+	+	+	+
ACC deamination	*acdS*	+	+	−	−	−	+	+
Petrobactin siderophore	*aslA*	+	+	+	+	+	+	+
Nitrogen fixation	*glnG*	+	+	+	+	+	+	+
	*glnL*	+	+	+	+	+	+	+
	*cheY*	+	+	+	+	+	+	+
	*cfbC*	+	+	+	+	+	+	+
	*glnK*	+	+	+	+	+	+	+
	*ntrX*	+	+	+	+	+	+	+
	*fixH*	+	+	+	+	+	+	+
	*nifT*	+	+	+	+	+	+	+
	*nifN*	+	+	+	+	+	+	+
	*ntrB*	+	+	+	+	+	+	+
	*ntrY*	+	+	+	+	+	+	+
	*lysR*	+	+	+	+	+	+	+
	*nifB*	+	+	+	+	+	+	+
	*nifE*	+	+	+	+	+	+	+
	*fdxB*	+	+	+	+	+	+	+
	*nifD*	+	+	+	+	+	+	+
	*nifH*	+	+	+	+	+	+	+
	*nifK*	+	+	+	+	+	+	+

## Data Availability

Data is contained within the article and Appendix A.

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
