# Peer review of "Genomic and Metabolic Characterization of Plant Growth-Promoting Rhizobacteria Isolated from Nodules of Clovers Grown in Non-Farmed Soil"

_ijms, 2023, doi:10.3390/ijms242316679_

Round 1

Reviewer 1 Report

Comments and Suggestions for Authors

Authors have isolated rhizobium strains from two types of clovers growing in agriculture and nonfarmed soil. Genome of 5 strains were sequenced. However metabolic profiling and PGP assays were performed for all strains.

Manuscript is well written; however, it needs further improvement. 

1. Title may state Rhizobium strains rather than PGPR, as whole manuscript is about rhizobium.

2. Abstract. it is worth mentioning that how many strains were isolated in total and characterized rather than using general statement.

3. Keywords. there is repetition. remove PGP, and PGPR.

4. Introduction is un-necessarily long.

    Line 65. Direct mechanisms include biofertlization....Which mechanism is

                  this?

   Last paragraph stating objectives of the study should end at Line 113. Rest

    of the lines are talking about methods used for this study. these need to

     be removed.

5. Results:  Line 150. What is BNP?

              Figure 1. What do they mean by "Bacteria"? if only that part of graph

               is bacteria then what about rest of the components, Acidobacteria,

                 Proteobacteria,etc,.etc,. are these not bacteria? it's confusing.

               Line 206. How was CAS method used for copper, aluminium and Arsenic? wat amendments were used in CAS medium?

Fig. 4. Y axis. is it really in mg or grams, kindly recheck. considering that it is fresh weight not dry weight 0.02 mg to 0.o4 mg weight of 6 weeks old plant is very low.

Footnote of figure 4 should state that data is average of 30 replicates.

What are the control strains? source of isolation? Why were 4 strains used as control?

6. Methods. Line 497. reference for primer sequences and PCR conditions are not provided.

what is 79CA medium/ provide teh reference when mentioned first time.

Line 536. "striking" is not the right word.

Line 543. PCR conditions for nodA gene amplification are nor provided.

Line 558. which medium? it only states liquid medium.

Line 567. How many strains were used for this experiment and which one?

Line 636. reference for M1 minimal medium is not provided.

Lines 647 to 649 need to be deleted.

Line 655. it is not mentioned that it was used for 4 metals. neither it is mentioned that any change was made for each metal.

7. Conclusion. lines 664 to 669 need to be removed. its conclusion not discussion.

8. References. Scientific names need to be italicized.

Reviewer 2 Report

Comments and Suggestions for Authors

The paper is very rich and of high scientific values. There are several suggestions for revision for the authors' reference.

1. In terms of materials and methods, when exactly was the sample collected? The basis for selecting this stage needs to be further supplemented.

2. Please summarize the key scientific issues of this paper.

3. In the discussion, please focus on several topics. The current layout is not very readable.

4. How can the results obtained in this paper be applied in the future? Appropriate outlook is recommended.

Reviewer 3 Report

Comments and Suggestions for Authors

In this manuscript, authors showed the “Genomic and metabolic characterization of plant growth-promoting rhizobacteria isolated from nodules of clovers grown in non-farmed soil” The manuscript may have potential interest. However, the data showed in the manuscript was not with a high quality. The authors did a lot of NGS sequencing and analysis, but I think the main conclusions of the manuscript are missing. The authors must address several major comments before the consideration of publishing in Int. J. Mol. Sci.

1. I think the main conclusions of the manuscript are missing. The authors showed a lot of work about 16s and genomic sequencing. However, I think the authors should make it clear what the main conclusions are from these sequencing results. The authors showed that the rhizosphere soil microbiota of clover plants grown in non-farmed soil and agricultural soil didn’t exhibit significant difference, does that mean agricultural practices don’t have a significant effect on the composition of the rhizosphere microbiota?

For the KEGG enrichment analysis, the authors have many discussions in the Section 3. But I think the authors should add the key conclusions from these KEGG enrichment analysis in the Results part. I think one of the highlights of this manuscript is to explore how rhizosphere microbiota promote the plant growth through the metabolic pathways. So, the authors should clearly show the conclusions from the analysis in the Results part.

2. Some Figures are not with a high resolution, such as Figures 2 and 8. The authors should update and increase the figures quality.

3. For the data showed in the Figure 3, it is hard to say what the yellow halos surrounding the bacterial colonies are. I can just see halos surrounding the bacterial colonies for KB8, but I don’t think they are yellow. For KB7 and KB12, it is hard to say that there were yellow halos surrounding the bacterial colonies. So, I don’t think the results from lines 208-211 can convince me. The authors should show high quality data for figure 3.

4. The data in Figure 4 is very confusing. What are the letters of the bars? How does the letters represent the significant differences of the data in the bars? Does NC mean non-inoculated plants? Also, for lines 220-221, is KB3 not significantly higher than that of non-inoculated plants?

5. Lines 228-229, since the data in Figure 3 and 4 is confusing, I can’t get the reasons to select these five isolates for complete genome sequencing and comparative genomics.

6. I would suggest put Figure 5 into SI. I don’t think the audiences want to check the details of the genome in Figure 5.

7. Lines 286-292, the authors showed KB and KC genome groups were selected and subjected to functional enrichment for COG and KEGG pathways, but there were no results showed from COG enrichment analysis. If the authors didn’t plan to show those results, the COG should not be mentioned.

8. There are many abbreviations with no clear meaning.

What are the meanings of KB and KC?

Line 20, full name should be showed for ACC, since it was the first appearance.

Lines 127 and 139, if PGPB is plant growth-promoting taxa, it should be showed with the full name and abbreviation in the line 127 for the first time.

Line 207, what’s the full name of CAS?

Comments on the Quality of English Language

Minor editing of English language required

Round 2

Reviewer 3 Report

Comments and Suggestions for Authors

The authors have addressed most of my comments.

I just have one minor comment

For the letters in Figure 4 to represent the significant differences, I suggest the authors to add more details about this method in the figure legend. People who are not familiar with this approach to represent the significant differences may be confused.
